# Mechanochemical Approach to Obtaining a Multicomponent Fisetin Delivery System Improving Its Solubility and Biological Activity

**DOI:** 10.3390/ijms25073648

**Published:** 2024-03-25

**Authors:** Natalia Rosiak, Ewa Tykarska, Judyta Cielecka-Piontek

**Affiliations:** 1Department of Pharmacognosy and Biomaterials, Faculty of Pharmacy, Poznan University of Medical Sciences, 3 Rokietnicka St., 60-806 Poznan, Poland; nrosiak@ump.edu.pl; 2Department of Chemical Technology of Drugs, Poznan University of Medical Sciences, 3 Rokietnicka St., 60-806 Poznan, Poland; etykarsk@ump.edu.pl

**Keywords:** fisetin, amorphous solid dispersion, amorphous solid inclusion, miscibility, solubility, improved biological activity, molecular docking

## Abstract

In this study, binary amorphous solid dispersions (ASDs, fisetin-Eudragit^®^) and ternary amorphous solid inclusions (ASIs, fisetin-Eudragit^®^-HP-β-cyclodextrin) of fisetin (FIS) were prepared by the mechanochemical method without solvent. The amorphous nature of FIS in ASDs and ASIs was confirmed using XRPD (X-ray powder diffraction). DSC (Differential scanning calorimetry) confirmed full miscibility of multicomponent delivery systems. FT-IR (Fourier-transform infrared analysis) confirmed interactions that stabilize FIS’s amorphous state and identified the functional groups involved. The study culminated in evaluating the impact of amorphization on water solubility and conducting in vitro antioxidant assays: 2,2-azino-bis(3-ethylbenzothiazoline-6-sulfonic acid)—ABTS, 2,2-diphenyl-1-picrylhydrazyl—DPPH, Cupric Reducing Antioxidant Capacity—CUPRAC, and Ferric Reducing Antioxidant Power—FRAP and in vitro neuroprotective assays: inhibition of acetylcholinesterase—AChE and butyrylcholinesterase—BChE. In addition, molecular docking allowed for the determination of possible bonds and interactions between FIS and the mentioned above enzymes. The best preparation turned out to be ASI_30_EPO (ASD fisetin-Eudragit^®^ containing 30% FIS in combination with HP-β-cyclodextrin), which showed an improvement in apparent solubility (126.5 ± 0.1 µg∙mL^−1^) and antioxidant properties (ABTS: IC_50_ = 10.25 µg∙mL^−1^, DPPH: IC_50_ = 27.69 µg∙mL^−1^, CUPRAC: IC_0.5_ = 9.52 µg∙mL^−1^, FRAP: IC_0.5_ = 8.56 µg∙mL^−1^) and neuroprotective properties (inhibition AChE: 39.91%, and BChE: 42.62%).

## 1. Introduction

Fisetin (FIS) exhibits diverse properties, serving as an antioxidant, anti-inflammatory, antiallergic, neuroprotective, antidiabetic, anticancer, cardioprotective, and antiaging agent [1].

The pharmacological effects of FIS have been confirmed in recent years using in vitro and in vivo test models [2,3,4,5], offering promising treatment options for various neurological conditions [6]. FIS exerts its unique senolytic impact in the central nervous system by selectively targeting anti-apoptotic proteins. Through the inhibition of these proteins, FIS sensitizes senescent cells to programmed cell death, ultimately eliminating them. The literature confirms that the positive effects of FIS in neurological disorder models have been assessed in both preclinical and clinical studies [7,8,9,10,11,12,13,14,15]. Numerous preclinical investigations have shown that FIS may offer advantages in addressing neurological issues and neurodegenerative conditions such as Huntington’s, Parkinson’s, and Alzheimer’s diseases. Ahmad et al. [16] confirmed FIS’s ability to rescue brain against oxidative stress, neuroinflammation, and memory impairment in mice model. According to Ding et al. [17], in diseases like sepsis-associated encephalopathy, FIS reduces neuroinflammation, stimulates mitophagy, and improves cognitive impairment in rats. Clinical trials using medicines containing FIS registered in recent years in ClinicalTrials.gov (www.clinicaltrials.gov, accessed on 28 February 2024) pertained to, among others, mild cognitive impairment (NCT02741804), frailty childhood cancer (NCT04733534), coronavirus infection (including COVID-19, NCT04476953, NCT04771611, NCT04537299), improving skeletal health in older humans (NCT04313634), frail elderly syndrome (NCT03675724), and osteoarthritis (NCT04815902, NCT04210986). Furthermore, in clinical practice since 2016, vision-specific medical food containing FIS has been used [1]. This food supports and protects the function of mitochondria in the optic nerve cells of patients with glaucoma.

Despite the considerable therapeutic properties that FIS offers, its clinical application is hampered by significantly low bioavailability (44%) due to limited water solubility of ~10.45 µg∙mL^−1^ and poor absorption [18,19]. Consequently, the administration of FIS in oral and dermal dosage forms is constrained. For this reason, in recent years, investigators have pursued diverse strategies to augment FIS solubility with the goal of optimizing its delivery and effectiveness [20]. Studies investigated micelles, nanoparticles [21,22,23], amorphous solid dispersion [24], nanoemulsions [4,25,26], hydrogels [27,28,29], liposomes [30,31,32], nanocrystals [33], inclusion complexes with β-cyclodextrin (βCD) [34,35,36,37], 2-hydroxypropyl-β-cyclodextrin (HPβCD) [37,38], and γ-cyclodextrin (γCD) [35].

One of the methods to improve polyphenol solubility and, consequently, bioavailability is to obtain amorphous solid dispersions (ASDs). In ASDs, polyphenol is distributed inside a polymeric matrix. The amorphous state, compared to the crystalline phase, is usually characterized by a higher dissolution rate and better release kinetics [39]. Researchers are continuously exploring new polymer excipients or combinations of polymers to optimize polyphenol-polymer compatibility, stability, and performance in ASDs. This involves understanding the interaction between the polyphenol and polymer at the molecular level to achieve the desired dissolution behavior and stability [40]. The preparation of ASDs of polyphenols involves various techniques, including spray drying [41], ball milling [42,43], cryomilling [44], hot-melt extrusion [45], and freeze-drying [46]. To date, it has been possible to obtain ASDs of such polyphenols as: apigenin [47], baicalein [48], chrysin [49], daidzein [50], FIS [24], genistein [51], kaempferol [52], naringenin [53], pterostilbene [43], quercetin [54,55,56], resveratrol [57]. Our previous research on FIS also focused on obtaining ASDs via co-precipitation in a supercritical carbon dioxide environment [24]. The obtained results confirmed that amorphous FIS can modulate the gut microbiome and exhibit higher solubility and biological activity. Encouraged by the results obtained in improving the solubility of FIS, we continued research on the preparation of its systems based on another method, i.e., a mechanochemical method that does not require organic solvents. In the current study, we used a dual-carrier system to obtain a multifunctional FIS delivery system.

Therefore, this work aimed to obtain low-cost binary and ternary FIS amorphous dispersions using the mechanochemical method, characterized by improved physical properties related to apparent solubility, as well as biological properties, including in vitro antioxidant and neuroprotective activity. This goal was achieved by using the following methods: X-ray powder diffraction—XRPD (determination the amorphous state of FIS), differential scanning calorimetry—DSC (confirmation of dispersions miscibility), Fourier-transform infrared spectroscopy—FT-IR (identification of interactions responsible for maintaining the amorphous state of FIS), high-performance liquid chromatography—HPLC (determination of FIS content in amorphous dispersions), in vitro antioxidant assays: ABTS (antioxidant assay: 2,2′-azino-bis(3-ethylbenzothiazoline-6-sulfonic acid), DPPH (2,2-diphenyl-1-picrylhydrazyl), CUPRAC (Cupric Reducing Antioxidant Capacity), and FRAP (Ferric Reducing Antioxidant Power), and in vitro neuroprotective assays: inhibition of acetylcholinesterase—AChE and butyrylcholinesterase—BChE. 

## 2. Results and Discussion

Binary amorphous solid dispersions (ASDs) and ternary amorphous solid inclusions (ASIs) were prepared to overcome fisetin (FIS) solubility limitations and improve its biological activity. Eudragit^®^ L100 (EL100) and Eudragit^®^ EPO (EPO) were chosen due to their high chemical stability and the ability to stabilize the amorphous form of polyphenols [52,57,58,59]. The use of ternary solid dispersion systems has represented an innovative approach to enhancing the solubility, wettability, and physical stability of amorphous drugs [60,61]. In our study, 2-hydroxypropyl-β-cyclodextrin (HPβCD) was loaded with ASDs of FIS. This cyclodextrin was selected due to its high solubility in water, exceptional compatibility with pharmaceutical formulations, and lack of toxicity [62,63]. HPβCD has a unique molecular structure resembling a truncated cone, inside which a hydrophobic drug (such as FIS) can be enclosed [64,65].

In our study, XRPD was used to differentiate between FIS amorphous and crystalline forms, and to confirm the formation of ASDs and ASIs. The diffraction pattern offers details about the atom arrangement within the solid, which can be utilized to determine whether the sample is crystalline or amorphous. Amorphous materials display a broad “halo” pattern, whereas crystalline materials show unique diffraction peaks (Bragg peaks) at particular angles.

The diffractograms of neat samples, FIS-EL100/FIS-EPO ASDs (ASD_L100/ASD_EPO) and FIS-EL100-HPβCD/FIS-EPO-HPβCD ASIs (ASI_L100/ASI_EPO) are shown in Figure 1.

Well-defined sharp peaks at 2Θ angles 7.9°, 10.8°, 11.6°, 12.5°, 14.1°, 15.5°, 17.5°, 21.8°, 24.1°, 25.6°, 26.3°, 28.2°, 31.3°, 32.4°, 34.8°, 37.0°, 40.5°, and 43.7° were visible in neat FIS. No Bragg peaks could be seen in the XRPD patterns of ASDs, ASIs, HPβCD, or the polymeric carrier EL100 and EPO, indicating that these materials are amorphous. The XRPD patterns of the binary physical mixtures are the superposition of FIS and Eudragit^®^ (Figure 1A,B). Both ASD-cyclodextrin ternary physical mixtures and ASIs were amorphous, as shown by “halo” effects in Figure 1C,D. However, the diffractograms of the PM and ASI with EL100 slightly differ in the nature of the spectrum, which is not the case when EPO is used.

It is the first time that ternary ASI of FIS with cyclodextrin was obtained by the mechanochemical method. 

Previously, the fully amorphous “halo” in X-ray patterns was observed in binary Eudragit^®^ systems reported by Wang et al. [59], Alsayad et al. [66], Chenchen et al. [67], and Zong et al. [68]. In another study, XRPD confirmed the amorphous state of curcumin [69], daidzein [50], genistein [51,70], hesperidin [42], pterostilbene [43,71], quercetin [41,72], and resveratrol [73] ASDs. 

Fatmi et al. [74] obtained a ternary inclusion complex of camptothecin:PEG 6000:cyclodextrins by solvent evaporation method. In another study, Mane et al. [75] prepared docetaxel:HPMC:βCD solid inclusion complexes via the freeze-drying method. Thiry et al. [76] used hot-melt extrusion (HME) to form itraconazole:Soluplus^®^:cyclodextrin ternary inclusion complexes. To date, inclusion complexes of FIS with βCD [34,35,36], HPβCD [38], and γcd [35] has been confirmed in the literature. 

Thermogravimetric (TG) analysis was conducted to assess the thermal stability of FIS, EL100, EPO, HPβCD, FIS-Eudragit^®^ ASDs, and ASIs. Figure 2 (black line) depicts a little weight loss of around 5% for FIS related to the evaporation of water from the crystalline sample [77]. FIS displayed a second significant mass loss (~30.6%) at a temperature of 369.6 °C, which could be due to the decomposition of the molecule. During the TG analysis conducted by Skiba et al. [78], similar results were observed (5% water loss and 32% mass loss around 360 °C).

The DSC thermogram of FIS (Figure 2, blue line) showed two endothermic peaks: the first at 135.1 °C confirming presence of water molecule, and the second effect at about 340.7 °C corresponding to the melting point (T_m_) of the crystalline form. Corina et al. [37] observed T_m_ of FIS at 348 °C, whereas Skiba et al. observed it at 330 °C [78]. A 30.6% mass loss of FIS at T_m_ indicates decomposition of the tested polyphenol. For this reason, amorphous FIS cannot be obtained by rapid cooling of the molten substance, and thus determine its glass transition temperature (T_g_). The decomposition of FIS at its T_m_ also occurred in ASDs and ASIs formulations (Appendix A). 

To the best of our knowledge, there is no information available in the literature concerning the preparation of amorphous pure FIS and determining its T_g_. Therefore, an attempt was made to obtain amorphous FIS using solvent evaporation and mechanochemical methods. XRPD analysis revealed that the techniques employed did not result in amorphization of FIS (Appendix A).

Since thermal events associated with moisture evaporation can mask the signals related to glass transition, the miscibility of ASDs and ASIs samples was measured in a heating–cooling–heating mode (see Section 3.4.3). During the second heating scan presented in Figure 3a,b, T_g_ values were observed during the second heating scan for raw EL100/EPO at 147.1/55.4 °C, while they were observed for ASD_20_EL100/ASD_20_EPO at 144.2/71.8 °C and for ASD_30_EL100/ASD_30_EPO at 145.9/76.7 °C. This indicates the presence of a single phase in all the ASDs and the miscibility of FIS and polymer in the specified ratios. T_g_ could not be observed for ASIs. 

Complete miscibility in compound Eudragit^®^ systems also were observed by Sathigari et al. (efavirenz-EPO) [79], Tian et al. (naproxen-EPO, ibuprofen-EPO) [80], and Liu et al. (indomethacin-EPO) [81]. Observed T_g_ values for pterostilbene-Soluplus^®^ systems [43], hesperidin-HPMC and hesperidin-Soluplus^®^ systems [42], sinapic acid-aminoacids [82], hesperidin-PVP solid dispersions [28], and curcumin-piperine-PVPVA 64 systems [69] have all been used to demonstrate full miscibility in other studies. 

As shown in Figure 3a, FIS-EL100 ASDs have T_g_ values lower than the polymer’s T_g_, which suggests that the FIS has a plasticizing effect on the EL100. Sathigari et al. [79] observed this effect for binary mixes of efavirenz and Plasdone S-630, and efavirenz and Eudragit^®^ EPO, while Kanaze et al. [83] observed it for hesperidin-PVP dispersions. According to the literature [83], the low molecular mass molecule may serve as a plasticizer, lowering the T_g_ value of the polymer, or it may interact weakly with the polymer to generate ASDs.

As shown in Figure 3b, the T_g_ value of FIS-EPO is greater than that of pure polymer for all ASDs, confirming the antiplasticization effect of FIS. Korhonen et al. [84] confirmed that the component with higher T_g_ value acted as an antiplasticizer of Eudragit^®^ EPO (increased the T_g_ of EPO). Chokshi et al. [81] observed the antiplasticization effect of the drug with EPO for indomethacin-EPO solid dispersions.

FT-IR analysis provides valuable insights into the structural properties of the analyzed material, allowing the identification of specific bonds and chemical interactions responsible for maintaining the amorphous state of the compounds in ASD. In this study, the FT-IR analysis aimed to identify distinctive peaks in the spectra of pure compounds (FIS, L100, EPO, HPβCD, see Figure 4) and assign specific molecular vibrations. Then, the obtained spectra were compared with ASDs and ASIs spectra to determine possible interactions between components. The characteristic absorption bands of FIS appear in the range of 400–1800 cm^−1^ (Figure 4, black line) and 3200–3600 cm^−1^ [24,85,86]. The assignments of the FIS bands were collected and compiled in Appendix A.

Eudragit^®^ is a copolymer of meta-acrylic acid containing numerous carboxylic groups. In the L100 form, the ratio of free to esterified carbonyl groups is 1:1. The IR spectrum of the ester and acid forms exhibits peaks at 1705 cm^−1^ and 1724 cm^−1^, respectively (see Figure 4, blue line). In addition, there are characteristic bands at 1153 cm^−1^ (–C–O–C stretching), 1192 cm^−1^ (C–O vibration of carboxylic acid), 1256 cm^−1^ (C–O vibration of carboxylic ester vibration), 1389 cm^−1^ (CH_x_), 1449 cm^−1^ (CH_3_), and 1481 cm^−1^ (CH_x_) [35]. The spectra of physical mixtures PM_20_EL100 and PM_30_EL100 (Figure 5a,b, grey line) are a composite of peaks from both FIS and EL100. The overlapping or combination of peaks and no changes in peak position in the spectra of physical mixtures indicate no interaction between the components. 

Meanwhile, for ASD_20_EL100 and ASD_30_EL100 (Figure 5a,b, blue line), there are noticeable differences in the spectral characteristics (summarized in Appendix A). The characteristic FIS bands corresponding to the HCCC bond (ring A, 808 cm^−1^, 872 cm^−1^, 974 cm^−1^), COH bond (ring A and B, 1117 cm^−1^, 1132 cm^−1^, 1437 cm^−1^), HCC bond (ring A and B, 1117 cm^−1^, 1437 cm^−1^, 1476 cm^−1^), CH (ring A and B, 3246 cm^−1^, 3346 cm^−1^), and OH (3518 cm^−1^, 3551 cm^−1^) disappeared. Moreover, the bands observed in the FIS spectrum at 627 cm^−1^ (OCCC and CCOC), 675 cm^−1^ (CCO), 770 cm^−1^ (CO), 854 cm^−1^ (HCCC), 1329 cm^−1^ (CC, COH, 3′–OH, 4′–OH), 1524 cm^−1^ (C–C) are visible at 621/621 cm^−1^, 673/672 cm^−1^, 775/773 cm^−1^, 847/847 cm^−1^, 1325/1325 cm^−1^, and 1508/1508 cm^−1^ for ASD_20_EL100 and ASD_30_EL100, respectively. Changes also pertain to the characteristic bands of EL100 that dominate in these spectra. The bands observed in the EL100 spectrum at 1153 cm^−1^ (–C–O–C stretching) and 1705 cm^−1^ (C–O stretching vibration of carboxylic ester) are visible at 1155/1155 cm^−1^ and 1699/1701 cm^−1^ for ASD_20_EL100 and ASD_30_EL100, respectively. Additionally, the band at 1481 cm^−1^ (CH_x_) disappeared, while the band at 1192 cm^−1^ (C–O vibration of carboxylic acid) increased in intensity. The bands at 1449 cm^−1^ (CH_3_) were indicative of shape change, and 1724 cm^−1^ (C=O stretching vibration in groups of carboxylic acids) decreased in intensity. Observed changes (shifting, intensity changes, and the disappearance of some absorption FIS and EL100 bands) confirmed the presence of interaction between FIS and EL100. 

The distinctive absorption bands of EPO are evident at 1144 cm^−1^ (C–N stretching of aliphatic amine and/or C–O stretching of ester), 1240 cm^−1^ (C–O stretching of ester), 1269 cm^−1^ (C–O stretching of ester), and 1454 cm^−1^ (C–H bending of methyl group) [35]. The spectra of physical mixtures PM_20_EPO and PM_30_EPO (Figure 6a,b, grey line) are a composite of peaks from both FIS and EL100. 

The overlapping or combination of peaks and no changes in peak position in the spectra of physical mixtures indicate no interaction between the components. Meanwhile, for ASD_20_EPO and ASD_30_EPO (Figure 6a,b, blue line), there are noticeable differences in the spectral characteristics (summarized in Appendix A).

The characteristic FIS bands corresponding to the HCCC bond (ring A, 808 cm^−1^, 872 cm^−1^, 974 cm^−1^), COH bond (ring A and B, 1132 cm^−1^, 1437 cm^−1^), HCC bond (ring A and B, 1437 cm^−1^, 1476 cm^−1^), CH (ring A and B, 3246 cm^−1^, 3346 cm^−1^), OH (3518 cm^−1^, 3551 cm^−1^) disappeared. Moreover, the bands observed in the FIS spectrum at 627 cm^−1^ (OCCC and CCOC), 675 cm^−1^ (CCO), 770 cm^−1^ (CO), 854 cm^−1^ (HCCC), 1018 (CCO and HCC), 1117 cm^−1^ (HCC, COH and 4′–OH), 1329 cm^−1^ (CC, COH, 3′–OH, and 4′–OH), 1524 cm^−1^ (C–C) are visible at 621/621 cm^−1^, 671/671 cm^−1^, 773/773 cm^−1^, 847/847 cm^−1^, 1015/1015 cm^−1^, 1120/1120 cm^−1^, 1325/1325 cm^−1^, and 1508/1508 cm^−1^ for ASD_20_EPO and ASD_30_EPO, respectively. Changes also pertain to the characteristic bands of EPO. The bands observed in the EPO spectrum at 1144 cm^−1^ and 1269 cm^−1^ are visible at 1146/1146 cm^−1^ and 1267/1267 for ASD_20_EPO and ASD_30_EPO, respectively. Additionally, the band at 1454 cm^−1^ (C–H bending of methyl group) increased in intensity and changed shape, whereas the band at 1240 decreased in intensity. Observed changes (shifting, intensity changes, disappearance of some absorption FIS and/or EPO bands) confirmed the presence of interaction between FIS and EPO.

It is indicated that the –C–O–C, –C=O, and/or –CH_3_ groups (carboxylic ester) of EL100, C–O, C=O, and/or C–H groups (carboxylic ester) of EPO may participate in interactions with the CCO, CO, HCCC, COH, and/or OH groups (ring A and/or B) of FIS. The obtained results indicate the presence of hydrogen bonds between FIS and Eudragit^®^. This is consistent with the literature. Previously, Sip et al. [24] confirmed that hydrogen bonds may form between the C–OH, C–O and/or –OH groups of FIS and the C=O and/or –OH groups of the polymer and maintain the amorphous state of FIS in FIS-copovidone ASD.

Molecular modeling was utilized to predict the existence of possible interaction in FIS-EL100 ASD and FIS-EPO ASD. After energy optimization at the B3LYP 6-31G’ level using Gaussian 16C, one hydrogen bond was identified between –OH group at the FIS A-ring/FIS B-ring and the carbonyl oxygen of one of the EL100/EPO ester groups (Figure 7a and Figure 7b, respectively). Theoretical predictions align well with the results of the FT-IR analysis.

The ATR-FTIR spectrum of HPβCD exhibited distinct absorption bands at 847 cm^−1^ corresponding to OH groups and the presence of glucopyranose units, 948 cm^−1^ associated with the presence of glucopyranose units, 1006 cm^−1^ representing C–H and C–O stretching vibrations, 1081 cm^−1^ reflecting wagging vibration of the C–H bonds and stretching vibrations of the C–C, C–O bonds, 1152 cm^−1^ related to C–O and C–H stretching vibrations. 

FT-IR data of physical mixtures PM_ASI_20_EL100/PM_ASI_20_EPO and PM_ASI_20_EL100/PM_ASI_30_EPO (Figure 8, grey line) are a composite of peaks from ASD and HPβCD, whereas for ASIs, the spectra have a characteristic HPβCD shape and confirmed that ASD is located inside the cyclodextrin (Figure 8, green line).

The band observed at 1006 cm^−1^ (HPβCD: C–H and C–O stretching vibrations) is also visible at 1028 cm^−1^ in ASI_EL100/ASI_EPO. It indicates hydrogen bond formation between the C–H and/or C–O group of HPβCD with the Eudragit^®^ group. The obtained results are consistent with the literature reports. Saokham et al. [87] observed shifts in the spectra of ternary systems and associated them with interactions between individual components. Al-Burtomani et al. [88] confirmed the formation of non-covalent bonds between the drug and cyclodextrin in the obtained ternary complexes.

The presence of a hydrophilic polymer (EL100, EPO) could facilitate the entry of FIS’s lipophilic groups into the hydrophobic interior of HPβCD. For this reason, the disappearance of the amorphous FIS bands that were visible in the ASD spectrum was observed in the ASI spectrum. Based on FT-IR results, it can be concluded that HPβCD can efficiently combine FIS-Eudragit^®^ ASD.

The enhanced solubility of polyphenol in the ASD is attributed to the reduced energy barrier necessary for molecule dissolution. In our research, the influence of FIS amorphization on its solubility was examined. It was confirmed that the pure FIS was insoluble in water. This is consistent with literature reports [18,19,24]. ASDs of FIS-EL100 and FIS-EPO did not improve solubility in water. In contrast, ASIs significantly improved solubility in this medium. The solubility of FIS within obtained systems in water decreased in the order of FIS30-EL100-HPβCD > FIS20-EL100-HPβCD > FIS30-EPO-HPβCD > FIS20-EPO-HPβCD, and the system of FIS-EL100-HPβCD (30% of FIS content) exhibited the maximum solubility, i.e., 318.3 ± 17.3 µg mL^−1^ (see Table 1).

The literature suggests that the best solubility improvement in water observed for ternary systems may be due to interactions between the components of ASIs in solution and molecular interaction-based solubilization of the drug’s amorphous form [89,90]. Increased solubility of drugs in amorphous form is associated with higher molecular mobility [91]. Amorphous drugs lack a well-defined crystalline structure, leading to higher molecular mobility compared to their crystalline forms. This increased mobility allows the drug molecules to interact more readily with the solvent molecules, thereby enhancing their solubility.

The literature reports that EPO can act as a solubilizing agent, improving the aqueous solubility of some BCS class II drugs [91]. In our study, Eudragits^®^ did not improve the water solubility of FIS. The same results were obtained in our previous work regarding kaempferol-Eudragit^®^ ASDs [52].

Literature reports have demonstrated that cyclodextrins are capable of encapsulating numerous lipophilic drugs within their central hydrophilic cavities, forming inclusion complexes in both aqueous solutions and solid states [92]. 

The formation of inclusion complexes is possible through the emergence of hydrophobic interactions, hydrogen bonding, or van der Waals forces [75,93].

Obtaining ternary systems of the active pharmaceutical ingredient API (API-polymer-cyclodextrin) is an increasingly common approach described in the literature to improve water solubility. Taupitz et al. [94] confirmed enhancing of itraconazole solubility by forming ternary systems with polymer (Soluplus^®^) in a dry complex with cyclodextrin (HPβCD) or hydroxybutenyl-β-cyclodextrin (HBen-β-CD). Ahad et al. [95] showed improved solubility of sinapic acid in water in the presence of HPβCD/HPMC. A similar observation was reported by El-Maradny et al. [89] for the meloxicam-HPβCD-PVP complexes.

According to the literature, the enhanced apparent solubility could have beneficial effects on the compound’s biological properties [24,42,43,46,96,97]. In our investigation, the increased solubility of FIS upon incorporation into ASIs resulted in alterations to its antioxidant and neuroprotective capabilities (see Table 2). 

FIS has a rotatable bond between the B-ring and C-ring, six hydrogen bond acceptors, and four hydrogen bond donors, and (see Figure 9) [1]. 

Studies on the structure-activity relationship (SAR) of FIS have been the subject of numerous literature reports [1,98,99,100,101]. It is indicated that the three-OH group has the greatest contribution to the antioxidant properties of FIS, while the double bond between the two- and three-carbon groups, and three′-OH and four′-OH groups (B-ring) are capable of enhancing its antioxidative activity [102]. Another study [103] concluded that the inhibitory activity against AChE is also influenced by the presence of the three-OH group (C-ring).

In our study, the antioxidant activity of ASDs and ASIs was analyzed using various methods, including ABTS—2,2′-azino-bis(3-ethylbenzothiazoline-6-sulfonic acid), DPPH—2,2-diphenyl-1-picrylhydrazyl, CUPRAC—Cupric Reducing Antioxidant Capacity, and FRAP—Ferric Reducing Antioxidant Power. The obtained results confirmed that the best system in regard to antioxidant properties was ASI_30_EPO (ABTS: 10.25 ± 0.24 µg∙mL^−1^, DPPH: 27.69 ± 1.96 µg∙mL^−1^, CUPRAC: 9.52 ± 0.03 µg∙mL^−1^, FRAP: 8.56 ± 0.07 µg∙mL^−1^).

Wang et al. [104] suggested that the hydrogen atom transfer (HAT) pathway may explain effective scavenges of DPPH radicals. They indicate that the 3′,4′-dihydroxyl moiety in the B ring of FIS plays an important role in this pathway (it could be ultimately oxidized to a stable ortho-benzoquinone form). However, in the case of ABTS, it is indicated that scavenging is based on the electron transfer reaction (SET) mechanism. Naeimi et al. based on bond dissociation energy (BDE) value, suggested a higher ability to donate H^+^ to free radicals by 3-OH (C-ring), 3′-OH (B-ring), and 4′-OH (B-ring) groups than the 7-OH of the A-ring [105].

Wang et al. also confirmed that in the case of the CUPRAC test, FIS’s efficiency in reducing Cu^2+^ to Cu^+^ is based on the SET mechanism. Firuzi et al. [91] indicate that the 2,3-double bond, 3-OH group, and 4-oxo functional group (C-ring), and the catechol group in the B ring give a major contribution to the reducing Fe^3+^ to Fe^2+^ capability in FRAP assay.

Rivera et al. [106] indicate that the strong antioxidant capacity of FIS contributes to regaining cellular redox equilibrium after ischemia, which may partly explain their neuroprotective potency. Sokal et al. suggest that FIS’s senotherapeutic and antioxidant properties can be the reason for its neuroprotective activity [107]. 

The inhibitory effect of FIS on cholinesterases AChE and BChE has been documented in the literature [24,108]. The SAR analysis suggests that the primary factor contributing to the highest acetylcholinesterase inhibitory activity in many flavonoids is the location and presence of –OH groups in the A and B rings, along with the unsaturation of ring C.

Raising FIS’s solubility may increase its bioavailability and enhance the potential to inhibit enzymes involved in neurodegeneration development. For this reason, in our study, in vitro tests were performed to determine the inhibitory effects of water solutions of FIS, its ASDs and ASIs on AChE and BChE inhibition. FIS exhibited a low inhibitory effect on AChE and BChE, with values of 0.40% ± 0.03% and 3.64% ± 0.23%, respectively. Improved solubility had a positive effect on the neuroprotective activity of FIS (see Table 2). The obtained results confirmed that the best system was ASI_30_EPO (AChE inhibition: 39.91 ± 3.47% and BChE inhibition: 42.62 ± 1.01%). Our results surpass those reported in earlier investigations involving a FIS-copovidone ASD (AChE inhibition: ~20% and BChE inhibition: ~30%) [24]. 

Molecular docking (MD) stands as a pivotal technique in structural molecular biology, a consensus affirmed by numerous citations in the literature [109,110,111,112,113,114]. In our study, MD was used to observe and confirm possible interactions between FIS and AChE/BChE. Figure 10a,c depict the active site gorges of AChE (PDB id: 4BDT) and BChE (PDB id: 4BDS) with docked FIS, respectively.

Hydrophobic interactions, hydrogen bonds, and π-stacking interactions were identified upon docking FIS to the AChE (binding energy: −11.71 kcal·mol^−1^) and BChE (binding energy: −10.40 kcal·mol^−1^) (Figure 10b,d, respectively). The lowest energy conformer of FIS-AChE showed hydrogen bonds with HIS^443^, SER^121^, THR^79^, and TYR^337^, hydrophobic interaction with ASP^70^, TRP^435^, and TYR^333^. FIS located parallel to TYR^333^ constitutes π–π stacking. The type of binding interactions between the FIS molecule and BChE was distinct from that between the FIS molecule and AChe because of a variation in the amino acid chain in the active site of BChE. In the BChE active site, FIS displayed hydrogen bonds with THR^117^, ASP^67^, and TYR^329^, hydrophobic interactions with TRP^79^ and ALA^325^, and π-stacking interaction with TRP^79^. The results are consistent with the literature [115,116,117]. For instance, Shi et al. [115] suggest that FIS is a substance that may have inhibitory effects on the AChE enzyme based on their theoretical and experimental research.

## 3. Materials and Methods

### 3.1. Materials

Reagents such as fisetin (FIS, purity > 95%), cyclodextrin (2-hydroxypropyl-β-cyclodextrin, HPβCD), ammonium acetate, the radicals necessary for the ABTS and DPPH assays, and FeCl_3_·6H_2_O, neocuproine, copper (II) chloride, sodium acetate trihydrate, and TPTZ, were purchased from Sigma Aldrich Chemie (Berlin, Germany). Eudragit^®^ L100 (EL100) and Eudragit^®^ EPO (EPO) were supplied by Röhm Pharma (Weiterstadt, Germany). HPLC grade methanol was obtained from Merck (Warsaw, Poland). The ultra-pure water necessary for HPLC and solubility tests was obtained thanks to the use of a water purification system (Millipore Direct-Q 3 UV, Merck, Darmstadt, Germany). Formic acid (85%) was provided by POCH (Gliwice, Poland). 

### 3.2. Preparation of Amorphous Solid Dispersion (ASD) of FIS

A physical mixture (500 mg) of FIS-EL100 and FIS-EPO (20% and 30% FIS content) was homogenized in a mortar. The milling process was carried out at room temperature according to the procedure proposed in our previous work [71]. Between tests, the prepared samples were stored in a desiccator.

### 3.3. Preparation of Amorphous Solid Inclusion (ASI) of FIS

Each ASD of FIS-Eudragit^®^ was grated in a mortar at a weight ratio of 1:1 with HPβCD. The physical mixture thus obtained was placed in a cylinder together with steel balls (3 pieces in diameter 12 mm). Before starting the grinding process, the cylinders were placed in the freezer (−20 °C) for 30 min to cool them down. The milling frequency was set to 30 Hz, and the milling time was set to 20 min. The obtained samples were stored protected from the influence of ambient conditions (in a desiccator).

### 3.4. Identification of Neat Compounds, ASDs and ASIs

#### 3.4.1. X-ray Powder Diffraction (XRPD) 

Powder X-ray diffractometry with a Bruker D2 Phaser diffractometer (Bruker, Germany) was employed to verify the physical state of the following samples: (i) compound: FIS, Eudragit L100, Eudragit EPO, HPβCD, (ii) the physical mixture, (iii) ASDs, and (iv) ASIs. The parameters of the device during measurement were consistent with the protocol presented in the previous work [71]. Diffractograms were recorded in the range of 5° to 40° 2Θ with a step size of 0.02° 2Θ and a counting rate of 2 s·step^−1^. The data were visualized using Origin 2021b (version 9.8.5.212, OriginLab Corporation, Northampton, MA, USA).

#### 3.4.2. Thermogravimetric Analysis (TG)

The thermal stability of FIS, EL100, EPO, ASDs, and ASIs was determined using a TG 209 F3 Tarsus^®^ micro-thermobalance (Netzsch, Selb, Germany). During the tests, 85 µL of open Al_2_O_3_ crucible was used. A crucible contained approximately 6–8 mg of the sample. TG measurements were carried out in the temperature range of 25–250 °C at a constant heating rate (10 °C per minute). The tests were conducted in a nitrogen atmosphere (flow rate 250 mL·min^−1^). Proteus 8.0 (Netzsch, Selb, Germany) and Origin 2021b (version 9.8.5.212, OriginLab Corporation, Northampton, MA, USA) were used to evaluate and visualize the collected data, respectively.

#### 3.4.3. Differential Scanning Calorimetry (DSC)

The registration of DSC thermograms was possible thanks to the use of a differential scanning calorimeter, model DSC 214 Polyma (Netzsch, Selb, Germany). Powder samples weighing 5–8 mg were put into sealed pans with a pinhole. The FIS melting point was recorded using the following measurement conditions: a single heating mode and a scanning rate of 10 °C·min^−1^. The glass transition temperature of (T_g_) of Eudragit^®^ EL100, Eudragit^®^ EPO, and ASDs was recorded using melting and cooling modes.

The optimized parameters of the melting (↑, for EL100 and EPO first heating: 10 °C·min^−1^ and second heating: 40 °C·min^−1^, whereas for ASDs first and second heating: 40 °C·min^−1^), cooling (↓, 40 °C·min^−1^ for all samples) and isotherm (→, 3 min for all samples) modes allowed the observation of the T_g_ of:
EL100: ↑30–250 °C; →250 °C; ↓250–10 °C; →10 °C; ↑10 °C–200 °C.EPO: ↑30–245 °C; →245 °C; ↓245–−10 °C; →−10 °C; ↑−10–120 °C.FIS-EL100 ASD: ↑30–245 °C; →245 °C; ↓245–10 °C; →10 °C; ↑10 °C–200 °C.FIS-EPO ASD: ↑5–120 °C; →120 °C; ↓120–5 °C; →5 °C; ↑5 °C–120 °C.

Measurements were carried out in a nitrogen atmosphere (250 mL per minute). DSC thermograms were analyzed using Proteus 8.0 (Netzsch, Selb, Germany). Origin 2021b (version 9.8.5.212, OriginLab Corporation, Northampton, MA, USA) was used to visualize the data.

#### 3.4.4. ATR-FTIR Spectroscopy

The registration of FT-IR spectra in the MIR range of 400–4000 cm^−1^ was made possible by a spectrophotometer IRTracer-100 with a QATR hold. Measurement parameters: resolution 4 cm^−1^, number of scans 100. Program for registered spectra: LabSolution IR software (version 1.86 SP2, Shimadzu, Kyoto, Japan). Origin 2021b (version 9.8.5.212, OriginLab Corporation, Northampton, MA, USA) was used to visualize the data.

#### 3.4.5. Molecular Modeling

The FIS molecular structure (in sdf format) was obtained from PubChem (PubChem CID: 5281614; website: https://pubchem.ncbi.nlm.nih.gov/, accessed on 1 March 2024). Gaussian 16C (Wallingford, CT, USA) software was utilized to optimize the FIS geometries (B3LYP/6-31 (d,p) level) prior to molecular modeling. Gaussview 6.0.16 was employed to draw the EL100 and EPO structures. Energy optimization of FIS-EL100 and FIS-EPO was conducted at the B3LYP 6-31G’ level using Gaussian 16C (Wallingford, CT, USA).

### 3.5. Studies of Results Introduction of FIS into ASD/ASI

#### 3.5.1. The Apparent Solubility

The apparent solubility of FIS was tested in 10 mL glass tubes containing an excess amount of FIS ASD and FIS ASI and 4 mL of water. Each sample was mixed for 30 s using a vortex mixer. Before HPLC testing, the samples were filtered (0.45 µm nylon membrane filter (Sigma-Aldrich, St. Louis, MO, USA)). The analysis was carried out three times. All HPLC assays were carried out using the Shimadzu Nexera (Shimadzu Corp., Kyoto, Japan). The process was carried out with the following measurement parameters: stationary phase: Dr. Maisch ReproSil-Pur Basic-C18 column (100 × 4.60 mm, 5 µm particle sizes) (Dr. Maisch, Ammerbuch-Entringen, Germany), mobile phase filtered via a 0.45 µm nylon filter: methanol and 0.1% trifluoracetic acid (50:50 *v/v*), injection volume 10 µL, flow rate 1 mL·min^−1^, column temperature 35 °C, and wavelength 366 nm.

#### 3.5.2. Antioxidant Properties

The antioxidant activity of the FIS, ASDs, and ASIs water solution was assessed using ABTS, DPPH, CUPRAC, and FRAP assays. The assays were performed in 96-well microplates. Measurements were made in a Multiskan GO UV reader (Thermo-Scientific, Waltham, MA, USA). The assays were conducted following previous protocols [81].

The ABTS and DPPH assays were conducted following the procedure reported earlier. Each sample (25 μL) was added to either the ABTS solution or DPPH solution (175 μL) and then incubated at room temperature for 30 min in the dark. After that, the absorbance was measured at λ = 517 nm (DPPH) and λ = 734 nm (ABTS) in comparison to a blank (DPPH: 25 μL of water + 175 μL of methanol, ABTS: 200 μL of water). The analysis made use of six replicates. Equation (1) was used to determine the percentage (%) of free radicals that could be removed.
(1)$$\mathrm{the}\;\mathrm{degree}\;\mathrm{of}\;\mathrm{radical}\;\mathrm{scavenging}\;(\%)=\frac{A_{0}-A_{i}}{A_{0}}*100\%$$
where A_0_ and A_i_ are the absorbance of the control and the sample, respectively.

The results were used to calculate the IC_50_ value, which represents the FIS concentration necessary to reduce radical generation by 50%.

During the CUPRAC assay, 50 µL of the sample’s aqueous solution and 150 µL of the CUPRAC solution were dispensed into the wells of a 96-well plate. The plate was then incubated at room temperature with agitation shielded from light for 30 min. Color alterations were measured at a wavelength of 450 nm.

During the FRAP assay procedure, 25 µL of aqueous solutions of samples and 175 µL of the FRAP solution (2.5 mL TPTZ solution, 25 mL acetate buffer, and 2.5 mL FeCl_3_·6H_2_O solution) were measured. The plate was incubated with shaking at 37 °C and protected from light for 30 min. Color changes were read at λ = 593 nm. 

The IC0.5 value, representing the concentration of the sample required to reach an absorbance of 0.5, was determined.

#### 3.5.3. Anticholinesterase Activity

Both AChE and BChE were inhibited using a test that was based on the spectrometric approach of Ellman et al. [82]. A spectrophotometric indication of enzyme activity is the increase in thiocholine’s color intensity on a 96-well plate. To each well, 25.0 µL of the test sample (a water solution containing FIS, FIS ASD, and FIS ASI obtained from the apparent solubility study) was added along with 30.0 µL of AChE/BChE solution at a concentration of 0.2 U·mL^−1^, and 40.0 µL of 0.05 M Tris-HCl buffer (pH 8.0). The plates were then incubated at room temperature for five minutes with agitation. Following this, each well received an additional 125.0 µL of 0.3 mM DTNB solution and 30.0 µL of 1.5 mM acetylthiocholine iodide/butyrylthiocholine iodide solution, followed by a 20-min incubation under the same conditions. The control sample contained 25.0 µL of water instead of the test sample. Blank samples were prepared by replacing AChE/BChE with TRIS-HCl buffer. The enzyme inhibition percentage was determined using the following formula:(2)$$\mathrm{AChE}/\mathrm{BChE}\;\mathrm{inhibition}\;(\%)=\frac{1-\left(A_{1}-A_{1b}\right)}{\left(A_{0}-A_{0b}\right)}\cdot 100\%$$
where: A_1_, A_1b_, A_0_, A_0b_—the absorbance of the test sample (1), blank of the test sample (1b), control (0), blank of the control (0b).

#### 3.5.4. Molecular Docking Study

Molecular docking studies determined possible interactions between FIS and AChE/BChE. PrankWeb (https://prankweb.cz/, accessed on 15 December 2023) was used to forecast the active sites for AChE and BChE [83,84,85]. Files in appropriate formats were prepared using OpenBabel 3.1.1. [86]. The preparation of AChE/BChE and FIS utilized MGLTools 1.5.6 with AutoDock 4.2 (ADT; Scripps Research Institute, La Jolla, San Diego, CA, USA) [87]. AutoDockVina 1.1.2. was used to carry out the actual molecular docking. PyMOL 2.5.1 (DeLano Scientific LLC, Palo Alto, CA, USA) [88] and the Protein-Ligand Interaction Profiler (PLIP server, https://plip-tool.biotec.tu-dresden.de/, accessed on 20 December 2023) [89] were used to display and characterize the structural interactions. 

FIS’s molecular structure was downloaded in sdf format from PubChem (CID: 5281614). Before the molecular docking process, the FIS geometries were optimized at the B3LYP/6-31G(d,p) basis set of DFT using the Gaussview software (version 6.0, Wallingford, CT, USA). Then, hydrogen atoms were added along with Gasteiger’s partial charge in AutoDock.

The structures of AChE (PDB code: 4BDT with 3.10 Å resolution) and BChE (PDB code: 4BDS with 2.10 Å resolution) were obtained from the Protein Data Bank (in pdb format). Following this, the ligands and water molecules in the AChE and BChE structures were removed. The files were then saved in pdb format and imported into the Chimera 1.16 software for structure repair. This process involved adding missing elements such as hydrogen atoms and any absent atoms [90]. The structures were automatically saved in mol2 format, then converted back to pdb format using OpenBabel 3.1.1. These prepared files were then re-imported into AutoDock Tools. The distance between the surface of the cholinesterases and the FIS molecule was constrained within a maximum radius of 0.375 Å. The grid box was centered around the active site pocket, as predicted by PrankWeb. As predicted by PrankWeb, the AchE active site contained TYR^68^, ASP^70^, TYR^73^, THR^79^, TRP^82^, ASN^83^, GLY^116^, GLY^117^, GLY^118^, TYR^120^, SER^121^, GLY^122^, LEU^126^, GLU^198^, SER^199^, TRP^282^, LEU^285^, SER^289^, PHE^291^, ARG^292^, PHE^293^, TYR^333^, PHE^334^, TYR^337^, TRP^435^, HIS^443^, GLY^444^, and TYR^445^. The BChE’s active pocket contained: ASP^67^, GLY^75^, SER^76^, TRP^79^, ASN^80^, GLY^112^, GLY^113^, GLY^114^, GLN^116^, THR^117^, GLY^118^, TYR^125^, GLU^194^, SER^195^, TRP^228^, PRO^282^, LEU^283^, SER^284^, VAL^285^, ALA^325^, PHE^326^, TYR^329^, PHE^393^, TRP^425^, HIS^433^, GLY^434^, and TYR^435^.

## 4. Conclusions

The mechanochemical method resulted in binary amorphous solid dispersions (ASDs, 20% and 30% of fisetin content in the Eudragit^®^ matrix) and ternary amorphous solid inclusions (ASIs, ASD+2-hydroxypropyl-β-cyclodextrin). FT-IR analysis supported by molecular modeling defined the hydrogen bonds between the C=O ester group of Eudragit^®^ L100 or EPO and the –OH group attached to the FIS A- or B-ring, respectively. The inclusion of ASD in the cyclodextrin interior contributed to the occurrence of additional intermolecular interactions, which additionally solubilized the obtained dispersions.

The obtained results confirmed increased solubility and biological properties of ASIs compared to prior studies using extraction with CO_2_ and copovidone as a polymer. These findings highlight the importance of exploring a variety of amorphization procedures to optimize the incorporation of bioactive compounds into the polymer matrix. Obtained results underscore the potential of amorphization in overcoming the challenges associated with the poor water solubility of FIS, thereby augmenting its therapeutic efficacy, particularly in the context of neurological disorders. 

Looking ahead, determining the appropriate amorphization method and polymer type and examining their impact on fisetin bioactivity offer promising opportunities to improve the therapeutic utility of this compound in various applications. The developed ternary fisetin delivery system, which improves solubility and enhances antioxidant and neuroprotective properties, could be the subject of further preclinical investigations, potentially leading to subsequent clinical studies.

## Figures and Tables

**Figure 1 ijms-25-03648-f001:**
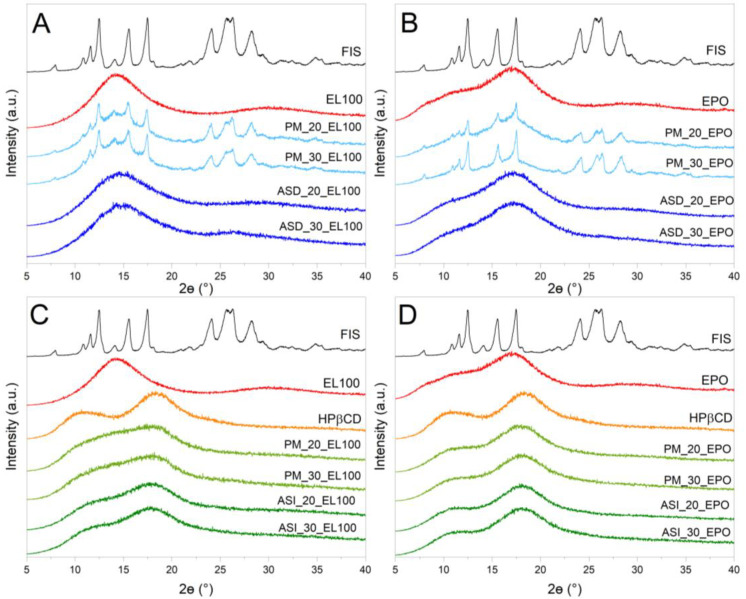
XRPD analysis: Diffractograms of neat compounds (fisetin—FIS, Eudragit^®^ L100—EL100, Eudragit^®^ EPO—EPO, 2-hydroxypropyl-β-cyclodextrin—HPβCD); (**A**) amorphous solid dispersion of FIS-EL100 (ASD), physical mixture of FIS-EL100 (PM); (**B**) amorphous solid dispersion of FIS-EPO (ASD), physical mixture of FIS-EPO (PM); (**C**) amorphous solid inclusion of FIS-EL100-HPβCD (ASI), physical mixture of FIS-EL100-HPβCD (PM); (**D**) amorphous solid inclusion of FIS-EPO-HPβCD (ASI), physical mixture of FIS-EPO-HPβCD (PM).

**Figure 2 ijms-25-03648-f002:**
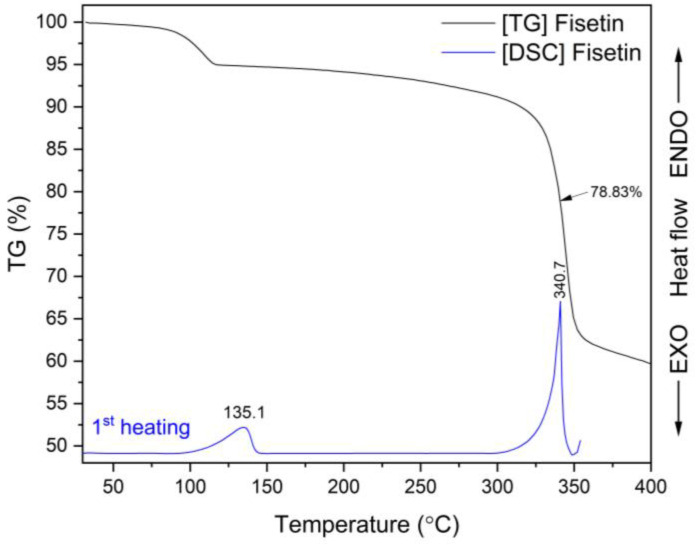
TG and DSC analysis: TG thermogram of fisetin (black line) and DSC thermogram recorded during the first heating scan for fisetin (blue line).

**Figure 3 ijms-25-03648-f003:**
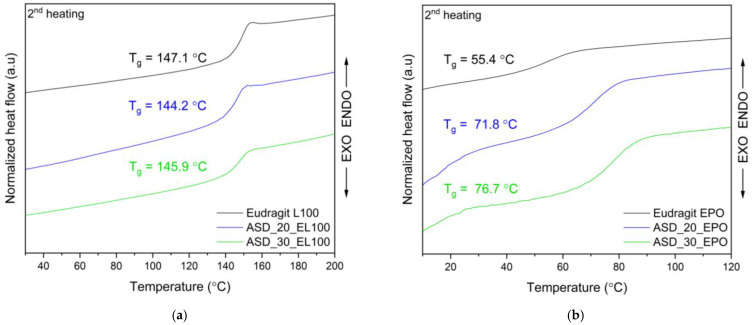
DSC analysis: (**a**) DSC thermograms (second heating scan) for Eudragit^®^ and amorphous solid dispersion of FIS-EL100 (ASD_EL100), T_g_—glass transition temperature; (**b**) DSC thermograms (second heating scan) for Eudragit^®^ EPO and amorphous solid dispersion of FIS-EPO (ASD_EPO), T_g_—glass transition temperature.

**Figure 4 ijms-25-03648-f004:**
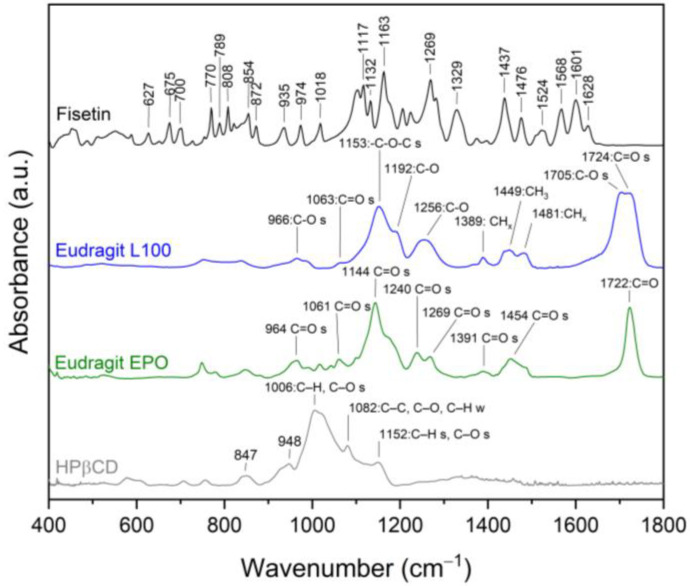
FT-IR analysis: fisetin, Eudragit^®^ L100, Eudragit^®^ EPO, 2-hydroxypropyl-β-cyclodextrin, range 400–1800 cm^−1^.

**Figure 5 ijms-25-03648-f005:**
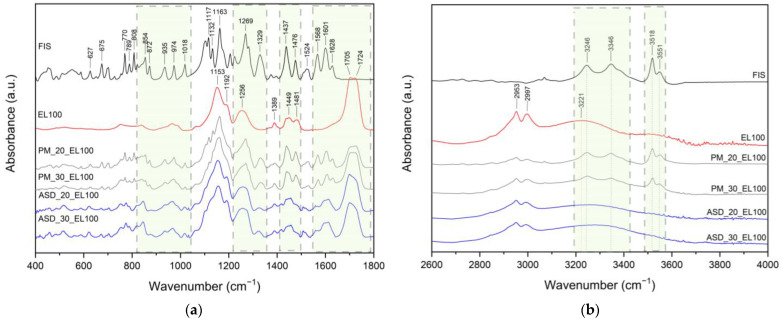
FT-IR analysis: fisetin (FIS), Eudragit^®^ L100 (EL100), physical mixture (PM), amorphous solid dispersion (ASD). (**a**) Range 400–1800 cm^−1^, (**b**) range 2600–4000 cm^−1^, box with gray dotted line—the most important changes in the FT-IR spectrum.

**Figure 6 ijms-25-03648-f006:**
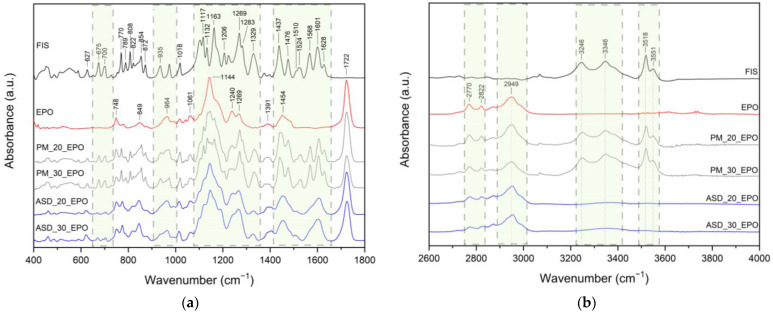
FT-IR analysis: fisetin (FIS, black line), Eudragit^®^ EPO (EPO, red line), physical mixture (PM, grey line), amorphous solid dispersion (ASD, blue line). (**a**) Range 400–1800 cm^−1^, (**b**) range 2600–4000 cm^−1^, box with gray dotted line—the most important changes in the FT-IR spectrum.

**Figure 7 ijms-25-03648-f007:**
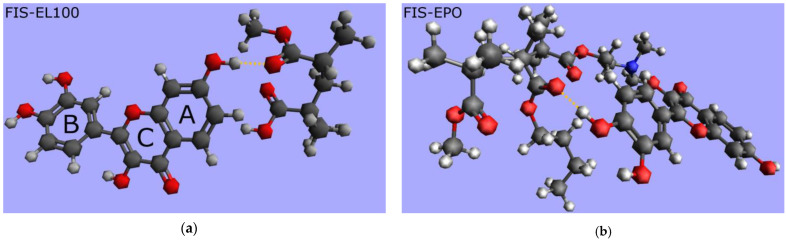
Molecular modelling of (**a**) fisetin-Eudragit^®^ L100 (FIS-EL100) and (**b**) fisetin-Eudragit^®^ EPO (FIS-EPO). Legend: orange dashed line—hydrogen bond.

**Figure 8 ijms-25-03648-f008:**
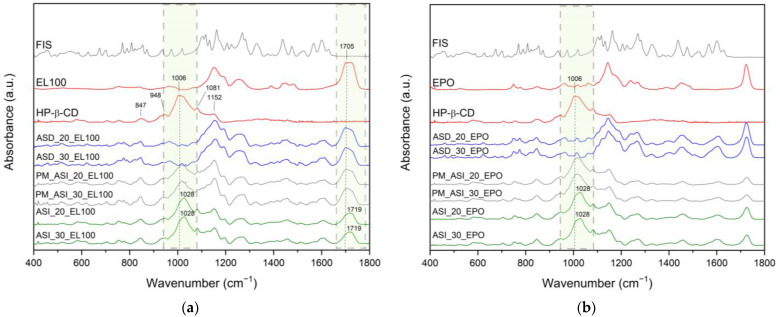
FT-IR analysis: (**a**) fisetin (FIS, black line), Eudragit^®^ L100 (EL100, red line), 2-hydroxypropyl-β-cyclodextrin (HPβCD, red line), physical mixture (PM, grey line), amorphous solid dispersion (ASD, blue line), amorphous solid inclusion (ASI, green line); (**b**) fisetin (FIS, black line), Eudragit^®^ EPO (EPO, red line), 2-hydroxypropyl-β-cyclodextrin (HPβCD, red line), physical mixture (PM, grey line), amorphous solid dispersion (ASD, blue line), amorphous solid inclusion (ASI, green line). Range 400–1800 cm^−1^, box with gray dotted line—the most important changes in the FT-IR spectrum.

**Figure 9 ijms-25-03648-f009:**
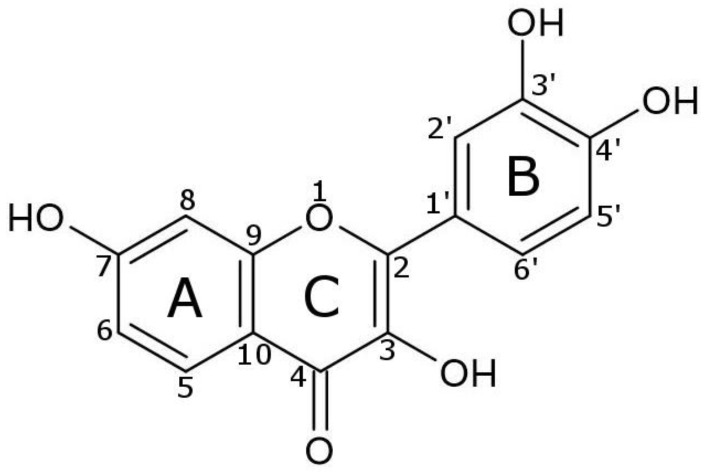
Chemical structure of fisetin (3,3′,4′,7-tetrahydroxyflavone).

**Figure 10 ijms-25-03648-f010:**
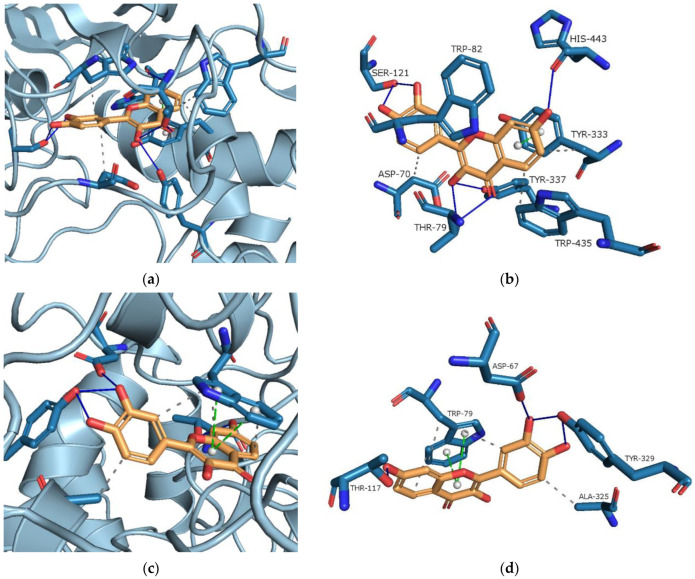
(**a**) The active site cavities of human acetylcholinesterase (AChE, PDB id: 4BDT); (**b**) binding orientation of FIS with AChE; (**c**) Active site cavities of human butyrylcholinesterase (BChE, PDB id: 4BDS); (**d**) Proposed binding orientation of FIS with BChE. The primary interactions of FIS with residues in the active sites of AChE and BChE consist of π-stacking interactions (shown with green dashed lines), hydrophobic interactions (represented by grey dashed lines), and hydrogen bonds (indicated by blue solid lines), orange structure—fisetin, blue structure—amino acid. Legend: ALA—alanine, ASP—aspartic acid, HIS—histidine, THR—threonine, TRP—tryptophan, TYR—tyrosine.

**Table 1 ijms-25-03648-t001:** Solubility (in water) of fisetin in amorphous solid dispersions (ASDs), and amorphous solid inclusions (ASIs) with carriers.

Content of FIS [%]	System	Concentration [µg mL^−1^]	Improved Solubility [–Fold]
20	ASD_20_EL100	-	none
	ASD_20_EPO	-	none
	ASI_20_EL100	221.7 ± 0.1	221.7
	ASI_20_EPO	77.5 ± 1.3	77.5
30	ASD_30_EL100	-	none
	ASD_30_EPO	-	none
	ASI_30_EL100	318.3 ± 17.3	318.3
	ASI_30_EPO	126.5 ± 0.1	126.5

Legend: FIS—fisetin, EL100—Eudragit^®^ L100, EPO—Eudragit^®^ EPO.

**Table 2 ijms-25-03648-t002:** Summarized of in vitro antioxidant and neuroprotective effects of amorphous solid inclusion (ASI) of FIS-EL100-HPβCD (ASI_20_EL100 and ASI_30_EL100) and FIS-EPO-HPβCD (ASI_20_ELPO and ASI_30_EPO).

Assay	Value		ASI_20_EPO	ASI_30_EPO	ASI_20_EL100	ASI_30_EL100
ABTS	IC_50_	[µg∙mL^−1^]	10.49 ± 0.70	10.25 ± 0.24	13.61 ± 0.51	15.23 ± 0.44
DPPH	IC_50_		27.52 ± 1.15	27.69 ± 1.96	33.20 ± 0.64	37.90 ± 0.73
CUPRAC	IC_0.5_		13.97 ± 0.67	9.52 ± 0.03	24.53 ± 0.30	28.56 ± 0.52
FRAP	IC_0.5_		13.05 ± 0.14	8.56 ± 0.07	22.37 ± 0.34	25.87 ± 0.26
AChE	AChE inhibition	%	22.74 ± 3.82	39.91 ± 3.47	19.43 ± 3.79	15.38 ± 3.43
BChE	BChE inhibition	%	41.37 ± 0.72	42.62 ± 1.01	27.93 ± 1.26	32.71 ± 1.58

Abbreviation: FIS—fisetin, EPO—Eudragit^®^ EPO, EL100—Eudragit^®^ L100, ABTS—2,2′-azino-bis(3-ethylbenzothiazoline-6-sulfonic acid) radical cation-based assay, DPPH—2,2-diphenyl-1-picrylhydrazyl assay, CUPRAC—cupric reducing antioxidant capacity assay, FRAP—ferric reducing ability of plasma assay, AChE—acetylcholinesterase, BChE—butyrylcholinesterase.

## Data Availability

The data are contained within the article and Appendix A.

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
