# Peer review of "Mechanochemical Approach to Obtaining a Multicomponent Fisetin Delivery System Improving Its Solubility and Biological Activity"

_ijms, 2024, doi:10.3390/ijms25073648_

Round 1
Reviewer 1 Report
Comments and Suggestions for Authors
Summary
The manuscript entitled ‘Mechanistic Approach to Obtaining a Muliticomponent Fisetin Delivery System Improving Its Solubility and Biological Activity’ presents the preparation of binary and ternary amorphous solid dispersions and amorphous solid inclusions, respectively.
The authors examined the potential of dry milling for creating amorphous systems with enhanced biological properties. Besides the experimental research, valuable molecular modeling was also applied for better understanding the interactions within the amorphous solid dispersion or inclusions. Although the manuscript contain interesting results some part of the work are not clarified enough.
For this reason, I would recommend the acceptance of this publication with major revision after consideration of some comments and remarks addressed.
General comments
1. The English writing should be further improved and correct linguistic mistakes of the manuscript.
2. The abbreviations should be used consequently during the whole manuscript. Here are some concrete examples:
- ‘ASD’ abbreviation is introduced in Line 53 in the main text but it is not used always after that. (It is similar in the case of ‘FIS’, ‘ASI’... etc.
- The ‘API’ abbreviation is not presented. Although it is a trivial abbreviation in the pharmaceutical field it would be better to define the first time it appears in the main text.
- ASD and ASI abbreviations are not presented in the abstract.
- It is unnecessary to use DSC and FT-IR abbreviations in the abstract.
- I am not sure that ‘ABTS, DPPH, CUPRAC...’ etc. abbreviations should be used in the abstract. It can be discussed later in the main text if it is necessary.
- Abbreviations of polymers, cyclodextrins, drugs are not defined all the time.
3. The first two paragraphs of the Introduction section can be merged and shortened.
4. I would suggest to use ‘mechanochemical approach’ in the title, and ‘mechanochemical method’ in Line 69 instead of ‘mechanistic’.
5. Line 166-167: Please use lower index for ‘g’ in the case of the Tg.
6. Figure 1 and 2: Please correct the ‘exo’ to ‘endo’ if you depict the thermograms this way. If exo arrow is pointing up than the water loss and the melting peak show a minimum.
7. Figure 1: It is suggested to depict the Heat Flow axis at the right side of the diagram.
Comments
1. I do not agree with this statement: “In the present study, ASD containing 20% and 30% FIS was obtained. The attained results suggest that dry milling could be a more efficient technique for incorporating FIS into the polymer matrix.”
In the previous, referenced study, different polymer was used. For this reason, I do not think so that the supercritical fluid technology and ball milling can be compared here. The applied polymer can strongly influence the stability of the amorphous drugs.
2. The results of Figure 2 should be clarified. So the problem was that the melting and the decomposition was too close to each other?
3. The evaluation of the FT-IR spectra could be more focused. Similar to Figure 7, only the main changes should be highlighted in Figure 5 and 6.
4. There are some sentences with references in the Results and discussion section, which would fit better into the introduction section.
5. Line 363-365: The sentence is not clear. Please reword it.
6. A table summarizing the prepared compositions can facilitate to follow the results.
Comments on the Quality of English LanguageThe English writing should be further improved.
Reviewer 2 Report
Comments and Suggestions for Authors
The manuscript "Mechanistic Approach to Obtaining a Multicomponent Fisetin Delivery System Improving Its Solubility and Biological Activity" by Natalia Rosiak, Ewa Tykarska, and Judyta Cielecka-Piontek explores the preparation of binary and ternary fisetin (FIS) delivery systems using the dry milling method. The study aims to evaluate the impact of amorphization on water solubility, antioxidant properties, and neuroprotective effects of FIS. The manuscript presents valuable insights into the development of fisetin delivery systems, addressing the crucial issue of low solubility. The scientific rigor demonstrated through comprehensive characterization and analysis is commendable. To enhance the manuscript's impact and novelty, further mechanistic insights into the observed enhancements and their implications for therapeutic applications should be provided. Addressing the suggested revisions will strengthen the manuscript and contribute significantly to the existing knowledge in the field of pharmaceutical sciences.
-The abstract provides a succinct overview of the study's objectives and methods employed to investigate the multicomponent delivery systems of fisetin (FIS). However, it could be improved by adding specific outcomes of the study, particularly concerning the enhancements observed in solubility and biological activities for the prepared systems.
- The introduction effectively contextualizes the significance of fisetin in therapeutic applications and addresses the challenges associated with its low solubility. However, it could be enhanced by succinctly summarizing the specific objectives of the current study, thereby providing clearer guidance for readers.
- The investigation of interactions between FIS and carrier polymers (Eudragit L100, Eudragit EPO) and cyclodextrin (HPβCD) through FT-IR analysis is a notable aspect of the study. However, additional spectroscopic techniques or computational methods could be suggested to further elucidate the nature of these interactions, especially regarding hydrogen bonding and its influence on solubility enhancement.
- The observed improvement in solubility for the amorphous solid inclusions (ASIs) compared to amorphous solid dispersions (ASDs) is a significant finding. However, it would be beneficial to include a more detailed discussion on the molecular mechanisms underlying this enhancement, including the role of carrier polymers and cyclodextrin in facilitating FIS solubilization.
-The study effectively compares the performance of binary (ASDs) and ternary (ASIs) systems. However, further analysis, such as a comparative assessment of other similar delivery systems for some drugs reported in the literature [10.1007/s40005-019-00434-2 10.1016/j.molliq.2022.119548 10.1002/9781118444726.ch3], could provide additional context and highlight the novelty and significance of the current findings. Please discuss these works in the text of the manuscript.
-The conclusion concisely summarizes the key findings of the study. However, it could be strengthened by highlighting the broader implications of the observed enhancements in solubility and biological activity for the potential clinical applications of fisetin-based therapies.
Implementing these revisions will significantly enhance the scientific quality and impact of the manuscript.
Round 2
Reviewer 1 Report
Comments and Suggestions for Authors
The reviewer thanks the authors for considering the issues raised in the first round of the review. The major issues raised by the reviewer are addressed well in the revised version of the manuscript. However, there are still some minor issues, which need to be corrected.
1. Please check to use everywhere the abbreviations after it was introduced at the first time in the main text. (Sometimes the whole drug name (‘fisetin’) or the whole formulation name (‘amorphous solid dispersion’) are written.)
2. Correct Figure 2.: Please add the unit of the Heat Flow axis, or use the (a.u.) abbreviation as arbitrary unit.
3. Line 378: Please use lower index for ‘m’ in the case of the Tm.
4. Line 407-411: Adding the Tg of pure fisetin would be more informative here. It is clear that the in situ ‘quench cooling’ in the DSC pan did not result the amorphous form of the fisetin but maybe it was possible by other methods (e.g. solvent methods). If the authors can support this section with some literature data, it would be fine.
Comments on the Quality of English LanguageSome typo can be corrected but it is fine.
